# Recruitment of Polo-like kinase couples synapsis to meiotic progression via inactivation of CHK-2

Liangyu Zhang[1,2,3,4], Weston T Stauffer[3,5], John S Wang[1,3], Fan Wu[1,3], Zhouliang Yu[1,2,3,4], Chenshu Liu[2,3], Hyung Jun Kim[1], Abby F Dernburg[1,2,3,4]*

[1]Department of Molecular and Cell Biology, University of California, Berkeley, Berkeley, United States; [2]California Institute for Quantitative Biosciences, Berkeley, United States; [3]Howard Hughes Medical Institute, Chevy Chase, United States; [4]Biological Systems and Engineering Division, Lawrence Berkeley National Laboratory, Berkeley, United States; [5]Department of Integrative Biology, University of California, Berkeley, Berkeley, United States

**Abstract** Meiotic chromosome segregation relies on synapsis and crossover (CO) recombination between homologous chromosomes. These processes require multiple steps that are coordinated by the meiotic cell cycle and monitored by surveillance mechanisms. In diverse species, failures in chromosome synapsis can trigger a cell cycle delay and/or lead to apoptosis. How this key step in 'homolog engagement' is sensed and transduced by meiotic cells is unknown. Here we report that in *C. elegans*, recruitment of the Polo-like kinase PLK-2 to the synaptonemal complex triggers phosphorylation and inactivation of CHK-2, an early meiotic kinase required for pairing, synapsis, and double-strand break (DSB) induction. Inactivation of CHK-2 terminates DSB formation and enables CO designation and cell cycle progression. These findings illuminate how meiotic cells ensure CO formation and accurate chromosome segregation.

*For correspondence: afdernburg@berkeley.edu

Competing interest: The authors declare that no competing interests exist.

## Editor's evaluation

Zhang et al. present convincing data describing a role for Polo-like kinase PLK-2 in restricting the activity of Chk2 kinase and coordinating synapsis of homologous chromosomes with the progression of meiotic prophase in *C. elegans*. By revealing PLK-2-dependent and -independent mechanisms of CHK-2 activity, this work provides a valuable understanding of the major regulators of meiotic progression.

## Introduction

Meiotic progression differs in many ways from a mitotic cell cycle. A single round of DNA replication at meiotic entry is followed by two nuclear divisions. Between replication and the first division is an extended period known as meiotic prophase, during which chromosomes pair, align through the process of synapsis, and recombine to establish physical links (chiasmata) between each pair of homologs. Together with sister chromatid cohesion, chiasmata direct bipolar orientation and segregation of homologous chromosomes during Meiosis I (*Hillers et al., 2017*; *Hunter, 2015*; *Ur and Corbett, 2021*). Synapsis and recombination are monitored by meiotic checkpoints that can delay cell cycle progression and/or lead to apoptosis when these processes are impaired (*Subramanian and Hochwagen, 2014*).

**eLife digest** Most animals, plants, and fungi reproduce sexually, meaning that the genetic information from two parents combines during fertilization to produce offspring. This parental genetic information is carried within the reproductive cells in the form of chromosomes. Reproductive cells in the ovaries or testes first multiply through normal cell division, but then go through a unique type of cell division called meiosis. During meiosis, pairs of chromosomes – the two copies inherited from each parent – must find each other and physically line up from one end to the other. As they align side-by-side with their partners, chromosomes also go through a mixing process called recombination, during which regions of one chromosome cross over to the paired chromosome to exchange information. Scientists are still working to understand how this process of chromosome alignment and crossing-over is controlled.

If chromosomes fail to line up or cross over during meiosis, eggs or sperm can end up with too many or too few chromosomes. If these faulty reproductive cells combine during fertilization this can lead to birth defects and developmental problems. To minimize this problem, reproductive cells have a quality control mechanism during meiosis called "crossover assurance", which limits how often mistakes occur.

Zhang et al. have investigated how cells can tell if their chromosomes have accomplished this as they undergo meiosis. They looked at egg cells of the roundworm *C. elegans*, whose meiotic processes are similar to those in humans. In *C. elegans*, a protein called CHK-2 regulates many of the early events during meiosis. During successful meiosis, CHK-2 is active for only a short amount of time. But if there are problems during recombination, CHK-2 stays active for longer and prevents the cell division from proceeding.

Zhang et al. uncovered another protein that affects for how long CHK-2 stays switched on. When chromosomes align with their partners, a protein called PLK-2 sticks to other proteins at the interface between the aligned chromosomes. A combination of microscopy and test tube experiments showed that when PLK-2 is bound to this specific location, it can turn off CHK-2. However, if the chromosome alignment fails, PLK-2 is not activated to switch off CHK-2. Therefore, CHK-2 is only switched off when the chromosomes are properly aligned and move on to the next step in crossing-over, which then allows meiosis to proceed. Thus, PLK-2 and CHK-2 work together to detect errors and to slow down meiosis if necessary.

Further experiments in mammalian reproductive cells will reveal how similar the crossover assurance mechanism is in different organisms. In the future, improved understanding of quality control during meiosis may eventually lead to improvements in assisted reproduction.

During early meiosis, double-strand breaks (DSBs) are induced and resected. These recombination intermediates undergo strand invasion to establish joint molecules between homologs, which recruit a set of factors collectively known as 'pro-crossover' or 'ZMM' proteins. Many of these intermediates eventually lose the ZMM proteins and are resolved as non-crossovers, while the subset that retains these factors is 'designated' to be resolved as crossovers (COs) (*Hunter, 2015*; *Pyatnitskaya et al., 2019*).

In *C. elegans*, formation and initial processing of DSBs requires the activity of CHK-2, a meiosis-specific homolog of the DNA damage transducing kinase Chk2/CHEK2 (*MacQueen and Villeneuve, 2001*; *Oishi et al., 2001*). CHK-2 is eventually inactivated at mid-pachytene; based on cytological markers, this roughly coincides with cessation of DSB formation and the resolution of many intermediates through a 'generic' (as opposed to meiosis-specific) homologous recombination pathway. A similar transition during mid-prophase has been described in budding yeast and mammals. In each case, this progression depends on assembly of the synaptonemal complex (SC), a meiosis-specific protein scaffold that assembles between homologous chromosomes (*Enguita-Marruedo et al., 2019*; *Hayashi et al., 2007*; *Kauppi et al., 2013*; *Lee et al., 2021*; *Mu et al., 2020*; *Murakami et al., 2020*; *Nadarajan et al., 2017*; *Thacker et al., 2014*; *Wojtasz et al., 2009*).

CHK-2 is inactive in proliferating germline stem cells and becomes active upon meiotic entry. The mechanism of activation has not been fully established, but it is promoted by targeted degradation of a CHK-2 inhibitor, PPM-1.D/Wip1 (*Baudrimont et al., 2022*). Work in budding yeast has shown that

a CHK-2 ortholog, Mek1, undergoes trans-autophosphorylation of its activation loop (*Carballo et al., 2008*; *Niu et al., 2007*); this mechanism is likely conserved in *C. elegans* since the activation loop of CHK-2 also contains a CHK-2 consensus phosphorylation motif (R-x-x-S/T) (*O'Neill et al., 2002*). Like all Chk2 homologs, CHK-2 contains a forkhead-associated (FHA) domain and thus recognizes phosphorylated motifs of the consensus (pT-X-X-[I/L/V]) (*Durocher and Jackson, 2002*; *Li et al., 2000*). While mammalian Chk2 undergoes homodimerization through binding of its FHA domain to N-terminal motifs phosphorylated by ATM (*Oliver et al., 2006*), the meiosis-specific Mek1 (*S. cerevisiae* and *S. pombe*) and CHK-2 (*C. elegans*) kinases lack this regulatory domain. Activation is likely promoted by binding of these kinases to target motifs on other proteins, such as Hop1 in budding yeast (*Carballo et al., 2008*), resulting in high local concentrations that enable intermolecular trans-phosphorylation. In *C. elegans*, the Pairing Center proteins HIM-8 and ZIM-1, -2, -3 contain FHA-binding motifs that recruit CHK-2 during early meiotic prophase. By phosphorylating a Polo box interacting motif on these proteins, CHK-2 primes the recruitment of the Polo-like kinase PLK-2 (*Harper et al., 2011*; *Kim et al., 2015*). Together, CHK-2 and PLK-2 at Pairing Centers modify NE proteins including lamin (LMN-1) and the LINC protein SUN-1, which promotes chromosome pairing and synapsis (*Link et al., 2018*; *Penkner et al., 2009*; *Sato et al., 2009*; *Woglar et al., 2013*). Thus, CHK-2 is required for chromosome pairing as well as for DSB induction.

During early pachytene, recombination intermediates are established along chromosomes and can be detected as foci by immunofluorescence. By mid-pachytene, most of these recombination intermediates disappear, while six 'designated crossover' foci, which show strong localization of the cyclin homolog COSA-1 and other pro-CO factors, are detected in each nucleus, one per chromosome pair (*Woglar and Villeneuve, 2018*; *Yokoo et al., 2012*; *Zhang et al., 2018*). This accumulation of pro-CO proteins at the sites that will eventually become crossovers is termed 'crossover designation'. Collectively, pro-CO factors are thought to prevent non-CO resolution and eventually promote the resolution of these late intermediates as COs. The mechanism and timing of CO resolution in *C. elegans* remain somewhat unclear.

Defects in synapsis or establishment of CO intermediates result in activation of a 'crossover assurance checkpoint' and a delay in cell cycle progression (reviewed in *Yu et al., 2016*). Activation of the checkpoint is detected cytologically as an extended region of CHK-2 activity. Based on nuclear morphology and cytological reporters of CHK-2 activity, defects in synapsis prolong the 'transition zone' (leptotene/zygotene) stage of meiosis, which is characterized by clustering of chromosomes and high CHK-2 activity, while defects in recombination result in a prolonged 'early pachytene' stage, with more dispersed chromosomes and intermediate level of CHK-2 activity (*Kim et al., 2015*). In mutants that are competent to form some COs despite defects in synapsis or recombination, the region of nuclei displaying a limited number of 'designated' CO sites is also shifted proximally within the germline (*Kim et al., 2015*; *Rosu et al., 2013*; *Stamper et al., 2013*; *Woglar and Villeneuve, 2018*; *Zhang et al., 2018*). This feedback regulation requires a family of HORMA domain proteins that bind to the chromosome axis (*Kim et al., 2015*), but how cells detect defects in synapsis or CO intermediates and how CHK-2 activity is prolonged remain unknown.

## Results
### CHK-2 inhibits CO designation

To monitor the activity of CHK-2 in *C. elegans* oocytes (*Figure 1A*), we used a phospho-specific antibody that recognizes CHK-2-dependent phosphorylation sites on the four paralogous Pairing Center proteins HIM-8 and ZIM-1, -2, and -3 (*Kim et al., 2015*). Co-staining of pHIM-8/ZIMs and GFP::CO-SA-1 confirmed that CO designation occurs concomitantly with loss of CHK-2 activity (*Figure 1B–D*). Since CHK-2 inactivation and CO designation are both delayed in mutants that activate the CO assurance checkpoint (*Kim et al., 2015*; *Zhang et al., 2018*), we wondered whether CHK-2 activity inhibits CO designation. Because CHK-2 activity during early meiotic prophase is essential for DSB formation and synapsis (*MacQueen and Villeneuve, 2001*; *Oishi et al., 2001*), and thus for the establishment of recombination intermediates that are prerequisite for CO designation, we exploited the auxin-inducible degradation (AID) system to deplete the CHK-2 protein (*Nishimura et al., 2009*; *Zhang et al., 2015*). Activity of degron-tagged CHK-2 became undetectable in meiotic nuclei within 3 hr of auxin treatment (*Figure 2—figure supplement 1A, B*). Depletion of CHK-2 shifted the appearance of

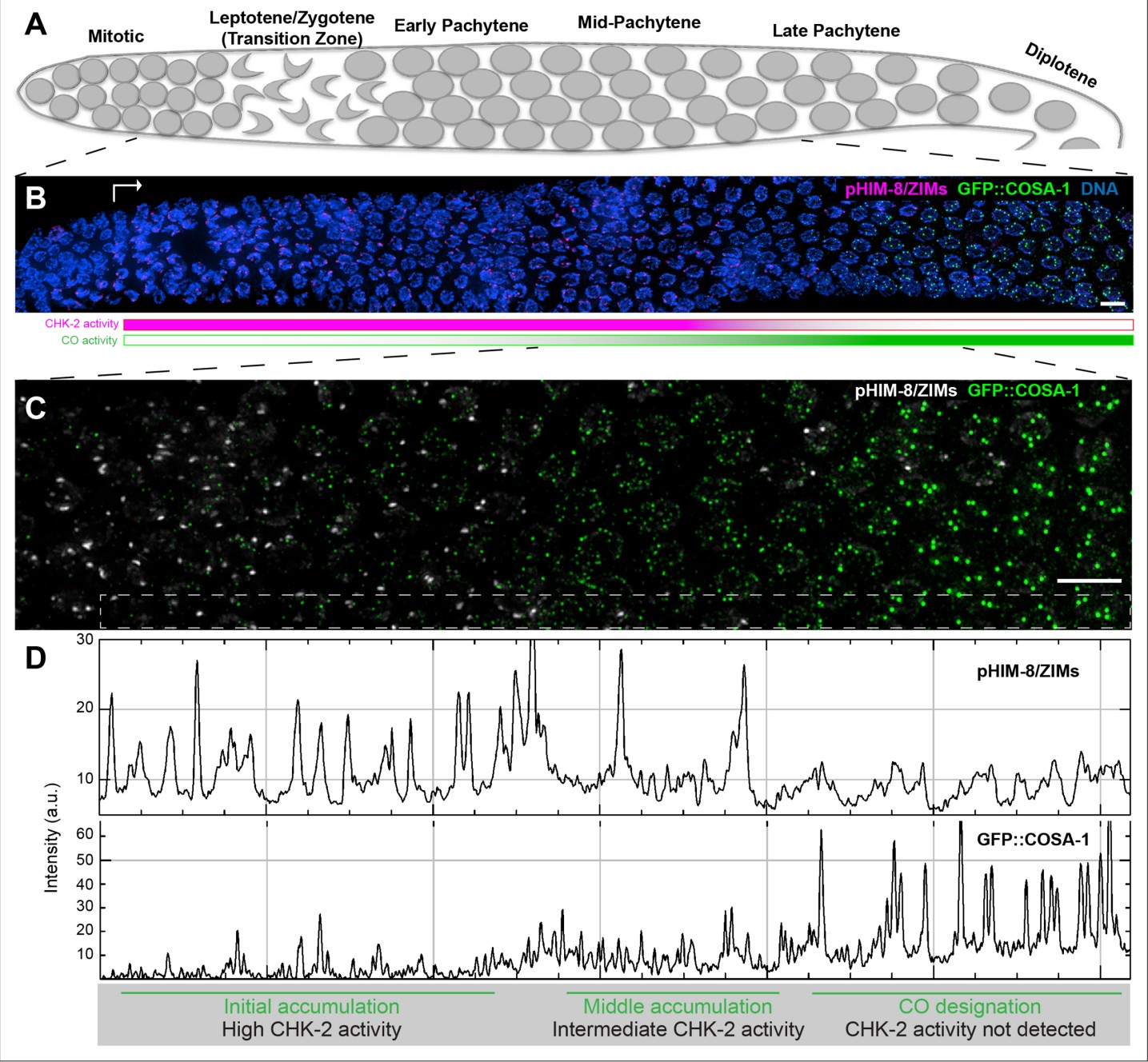

**Figure 1.** Temporal profile of CHK-2 activity and crossover designation during meiotic prophase in *C. elegans*. (**A**) Schematic of meiotic prophase in the *C. elegans* hermaphrodite germline. The distal tip of the gonad is oriented on the left side in this schematic and in all other figures. Meiotic progression is readily visualized due to the simple organization of the germline. Cells exit proliferation and enter meiosis in the distal 'arms' of the gonad and move proximally toward the spermatheca and uterus at a velocity of about one cell row per hour (***Deshong et al., 2014***). Homolog pairing and synapsis and double-strand break (DSB) induction normally occur during the first few hours following meiotic entry, in the 'transition zone' region corresponding to leptotene and zygotene. DSBs are then processed to form recombination intermediates, which increase in abundance during early pachytene. At mid-pachytene, a transition occurs that leads to disappearance of most intermediates. One recombination intermediate on each pair of chromosomes becomes 'designated' as an eventual crossover site. (**B**) Images of representative prophase nuclei from meiotic onset, labeled with markers for crossover intermediates (GFP::COSA-1, green) and CHK-2 activity (phospho-HIM-8 and ZIM-1/-2/-3, magenta). Bright GFP::COSA-1 foci are detected following inactivation of CHK-2. White arrow indicates meiotic onset. Scale bars, 5 μm. (**C**) Enlargement of the mid-pachytene nuclei shown in (**B**). phospho-HIM/ZIMs was pseudo colored to white for easy observation. Scale bars, 5 μm. (**D**) Line profiles indicating the relative fluorescence intensity of staining for phospho-HIM-8/ZIMs immunofluorescence (upper) and GFP::COSA-1 (lower) in the boxed region indicated in (**C**). Accumulation of pro-CO proteins can be stratified into three stages: an early stage with high CHK-2 activity and variable numbers of dim GFP::COSA-1; an intermediate stage with

*Figure 1 continued on next page*

Figure 1 continued

brighter and more abundant GFP::COSA-1 foci, and a post-designation stage, with a single CO-designated site per chromosome pair marked by bright GFP::COSA-1 fluorescence. The intensity of GFP::COSA-1 foci remains fairly constant throughout this last stage. In this work, we use the appearance of bright COSA-1 foci as a proxy for crossover designation and the mid-pachytene cell cycle transition.

nuclei containing bright COSA-1 foci to a more distal (earlier) position in the germline, indicative of accelerated CO designation and cell cycle progression (*Figure 2A, C*).

We reasoned that if CO designation is inhibited by CHK-2, then depletion of CHK-2 should be sufficient to restore earlier designation in mutants that disrupt establishment of CO intermediates on a subset of chromosomes. We depleted CHK-2 in *him-8* and *him-5* mutants, which specifically affect synapsis or DSB induction on the *X* chromosomes, respectively, but do not impair CO formation on autosomes (*Hodgkin et al., 1979*; *Broverman and Meneely, 1994*; *Phillips et al., 2005*; *Meneely et al., 2012*). In each case, we observed a shift in the appearance of bright COSA-1 foci toward the distal region of meiotic prophase following CHK-2 depletion (*Figure 2B, D*; *Figure 2—figure supplement 1C–F*). This result reinforces evidence described above that CHK-2 activity is required to delay CO designation in response to feedback regulation.

We observed some differences between the two mutants: in *him-8* mutants RAD-51 foci were far more abundant in the extended region of CHK-2 activity, and persisted even after CHK-2 was depleted, resulting in the appearance of nuclei with both RAD-51 foci and bright COSA-1 foci. In contrast, RAD-51 foci were sparser in *him-5* and mostly disappeared upon CHK-2 depletion (*Figure 2—figure supplement 1C–E*). These differences may reflect the different arrest points of the two mutants: *him-8* oocytes are defective in synapsis (*Phillips et al., 2005*), and thus arrest at a zygotene-like state with high CHK-2 activity, while *him-5* mutants are deficient in DSB initiation on the X chromosome (*Meneely et al., 2012*), and thus arrest at early pachytene with an intermediate level of CHK-2 activity. We speculate that progression to early pachytene and the associated reduction in CHK-2 activity in *him-5* mutants may attenuate break formation and/or allow breaks to progress to a more advanced stage of repair, so that upon CHK-2 depletion they are more rapidly resolved. Alternatively, the lower number of RAD-51 foci in *him-5* mutants may reflect a direct role for HIM-5 in promoting DSB formation (*Meneely et al., 2012*).

## Relocalization of Polo-like kinases to SCs promotes timely CHK-2 inactivation and CO designation

A key unanswered question is how CHK-2 is normally inactivated at mid-prophase to terminate DSB induction and promote CO designation. Prior work on inactivation of DNA damage response (DDR) signaling in proliferating cells revealed that the Polo-like kinase Plk1 enables mammalian cells to enter mitosis following DDR activation by phosphorylating the FHA domain of Chk2, which inhibits substrate binding (*van Vugt et al., 2010*). We wondered whether a similar mechanism might regulate CHK-2 in meiosis.

*C. elegans* expresses multiple homologs of mammalian Plk1, including PLK-1, which is essential for mitosis and thus for viability, and PLK-2, which is dispensable for development but plays important roles in meiosis (*Chase et al., 2000*; *Harper et al., 2011*; *Labella et al., 2011*). Loss-of-function mutations in *plk-2* perturb pairing and synapsis, and also cause a pronounced delay in CO designation and subsequent chromosome remodeling during late prophase (*Harper et al., 2011*; *Labella et al., 2011*). To test whether CHK-2 might be a substrate of PLK-2, we performed in vitro phosphorylation assays using purified PLK-2. We used purified kinase-dead CHK-2^KD as a substrate to avoid potential autophosphorylation of CHK-2. Mass spectrometry identified threonine 120 of CHK-2 as an in vitro target of PLK-2 (*Figure 3—figure supplement 1A*). This highly conserved residue lies within the FHA substrate recognition domain (*Figure 3A, B*), close to serine 116, which corresponds to a site phosphorylated by Plk1 in human cells during DNA damage checkpoint adaptation (*van Vugt et al., 2010*). S116 and T120 both conform to a consensus motif for Plk1 substrates (*Figure 3A*; *Santamaria et al., 2011*). These results indicated that PLK-2 might phosphorylate and inactivate CHK-2 during meiosis.

To determine whether CHK-2 is phosphorylated at Thr120 in vivo, we immunoprecipitated epitope-tagged CHK-2 from *C. elegans*. Transcriptome and proteome analyses have indicated that CHK-2 is very low in abundance (*Wang et al., 2015*; *Grün et al., 2014*). Consistent with this, tagged CHK-2 was difficult to detect on western blots from wild-type animals, even following immunoprecipitation.

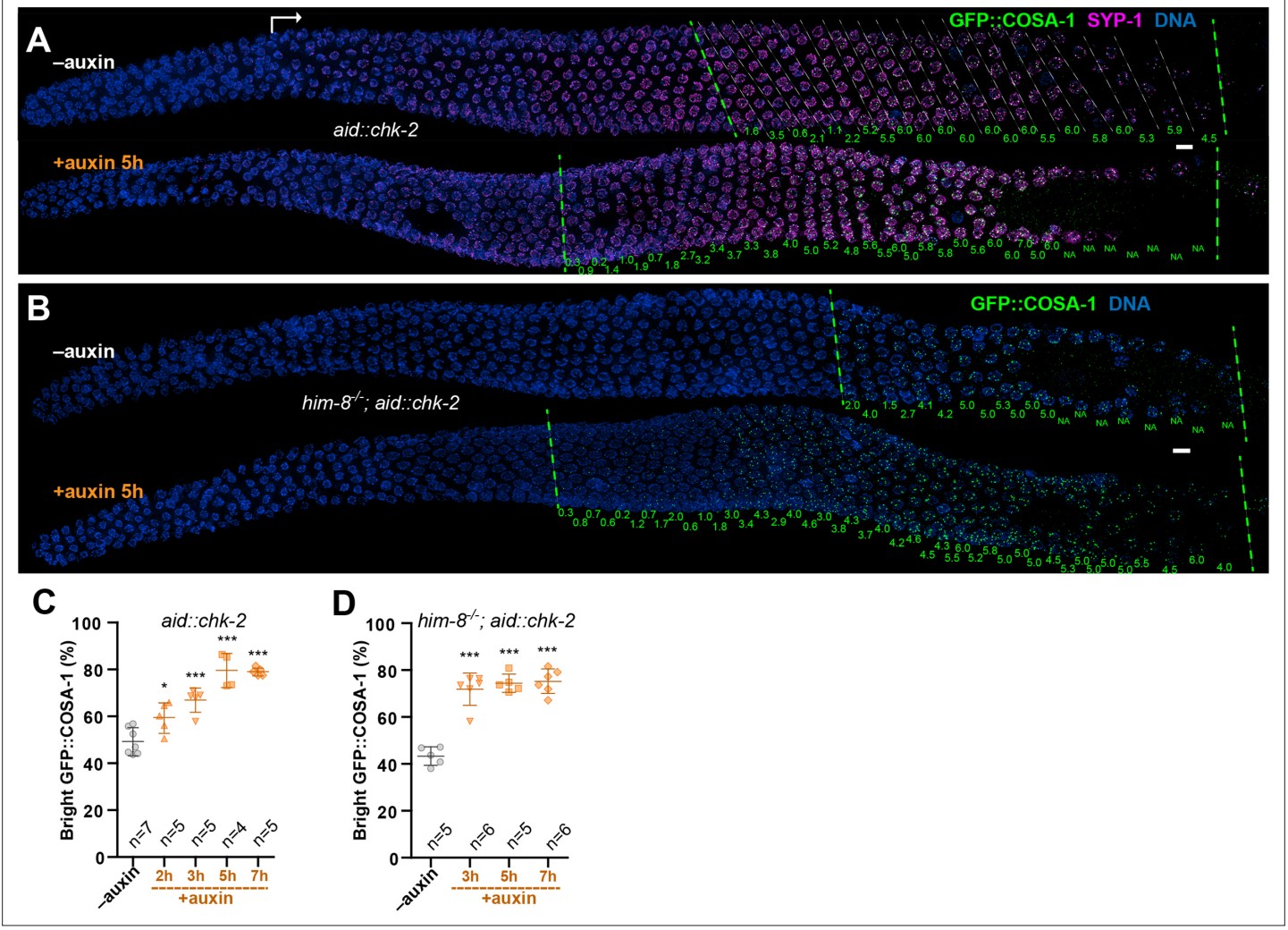

**Figure 2.** CHK-2 inhibits crossover (CO) designation. (**A**) Representative hermaphrodite gonads stained for GFP::COSA-1 (green), SYP-1 (magenta), and DNA (blue). Nuclei with bright COSA-1 foci are observed at a more distal position following CHK-2 depletion. Worms were exposed to 1 mM auxin (or 0.25% ethanol lacking auxin) for 5 hr before fixation. Dashed green lines on the left and right indicate the earliest nuclei with bright COSA-1 foci and the end of pachytene, respectively. White arrow indicates meiotic onset. The average number of bright COSA-1 foci per nucleus in each row is indicated below each image in green. Scale bars, 5 μm. (**B**) Germline from a *him-8* mutant hermaphrodite stained for GFP::COSA-1 (green), and DNA (blue), showing early appearance of bright COSA-1 foci upon CHK-2 depletion. Worms were treated with 1 mM auxin or solvent (0.25% ethanol) control for 5 hr before analysis. Scale bars, 5 μm. Note: the same image with RAD-51 is shown in *Figure 2—figure supplement 1C*. (**C**) CO designation occurs earlier upon CHK-2 depletion, as described in (**A**). Worms were exposed to auxin (or solvent control) for 2, 3, 5, or 7 hr before analysis. We define the 'Bright GFP::COSA-1 zone' as the length of the region from where bright GFP::COSA-1 foci appear to the end of pachytene, before oocytes form a single row of cells. Since the length of each meiotic stage region varies among individual animals, while the ratio between stages is relatively constant. We used the ratio of the length of 'Bright GFP::COSA-1 zone' to the length of the region from meiotic onset to the end of pachytene to reflect the timing of bright GFP::COSA-1 foci appearance and crossover designation. To simplify, we hereafter use 'bright GFP::COSA-1 (%)' in graphs to indicate this ratio. Meiotic onset was determined by the staining of meiosis-specific proteins SYP-1 and/or HTP-3. *n* = number of gonads scored for each condition. *p = 0.0161 and ***p = 0.0003, or <0.0001, respectively, two-sided Student's *t*-test. (**D**) Quantitative comparison of the timing of CO designation in worms as described in (**B**). Worms were treated with or without 1 mM auxin for 3, 5, or 7 hr before analysis. Quantification was performed as described in (**C**). ***p < 0.0001, two-sided Student's *t*-test.

The online version of this article includes the following figure supplement(s) for figure 2:

**Figure supplement 1.** Depletion of CHK-2 activity results in earlier crossover designation.

Based on other evidence (see below) we suspected that phosphorylation of CHK-2 might lead to its degradation. We thus exploited the AID system to deplete PAS-1, a subunit of the 20S proteasome, as a potential way to increase the abundance of CHK-2. Under conditions in which PAS-1 was efficiently depleted, CHK-2 was clearly enriched (*Figure 3—figure supplement 1B, C*). Mass

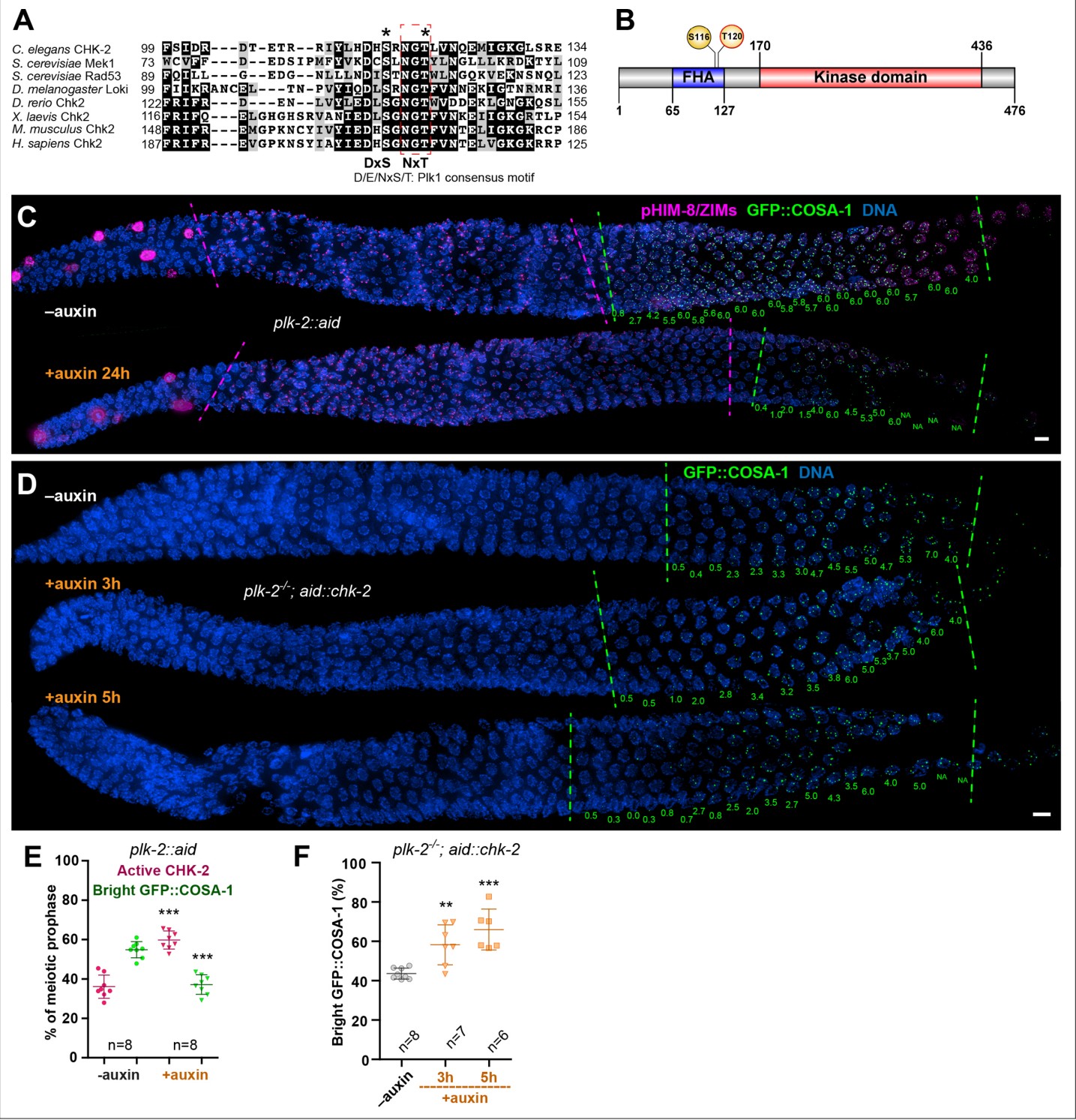

**Figure 3.** Inactivation of CHK-2 by Polo-like kinase promotes timely crossover (CO) designation. (**A**) Sequence alignment of CHK-2 orthologs from various eukaryotes generated with T-Coffee (**Notredame et al., 2000**), showing the conservation of Ser116 and Thr120 (asterisks). Thr120 is a direct target of PLK-2; Ser116 corresponds to a Plk1 site identified in mammlian cells (**van Vugt et al., 2010**). Black and gray shading indicate identical and similar residues, respectively. Both Ser116 and Thr120 match the Plk1 consensus motif [D/E/N]-X-[S/T] (**Santamaria et al., 2011**). (**B**) Schematic showing the domain organization of CHK-2 protein and the positions of two phosphorylation sites, Ser116 and Thr120. FHA: forkhead-associated domain. Numbers indicate amino acid positions. *C. elegans* CHK-2 and budding yeast Mek1 are meiosis-specific kinases that share the FHA and serine/threonine kinase domains of mammalian Chk2 and yeast Rad53, but lack the N-terminal SQ/TQ cluster that regulates activation of Chk2 by ATM. (**C**) Depletion of PLK-2 delays both CHK-2 inactivation and the appearance of bright COSA-1 foci. Worms were treated with 1 mM auxin for 24 hr and stained for pHIM-8/

*Figure 3 continued on next page*

*Figure 3 continued*

ZIMs (magenta), GFP::COSA-1 (green), and DNA (blue). Dashed magenta lines indicate the CHK-2-active zone. Green lines indicate the bright COSA-1 zone. The average number of bright COSA-1 foci per nuclei in each row is indicated in green below each image. Scale bars, 5 µm. (**D**) Depletion of CHK-2 restores early appearance of bright COSA-1 foci in *plk-2* mutants. Worms were treated with or without 1 mM auxin for 3 or 5 hr. Scale bars, 5 µm. (**E, F**) Quantification of the extension of the CHK-2-active zone and delay in appearance of bright COSA-1 foci in worms depleted for PLK-2, as described in (**C**) and of bright COSA-1 foci appearance in *plk-2* mutants upon depletion of CHK-2 as described in (**D**), respectively. *n* = number of gonads scored for each condition. **p = 0.0018 and ***p < 0.0001, two-sided Student's *t*-test.

The online version of this article includes the following source data and figure supplement(s) for figure 3:

**Figure supplement 1.** Phosphorylation of CHK-2 by PLK-2.

**Figure supplement 1—source data 1.** Western blotting raw images in *Figure 3—figure supplement 1*.

**Figure supplement 2.** Characterization of CHK-2 phospho-mutants.

**Figure supplement 2—source data 1.** Western blotting raw images in *Figure 3—figure supplement 2*.

spectrometry analyses of the immunoprecipitates revealed that CHK-2 is indeed phosphorylated on Thr120 (*Figure 3—figure supplement 1D*). To test the role of phosphorylation on the corresponding serine (S116) and/or T120 of CHK-2, we mutated each of these sites to nonphosphorylatable and phosphomimetic residues. Surprisingly, the abundance of all these mutant CHK-2 proteins was much lower than the wild-type protein; only S116A was detectable (*Figure 3—figure supplement 2A*). Consistent with the reduced abundance of CHK-2 protein, mutation of these residues led to loss of CHK-2 function: phopsho-HIM-8/ZIMs was not detectable and bivalent formation was defective; only S116A was partially functional (*Figure 3—figure supplement 2B*). These observations suggest that these mutations destabilize CHK-2 in addition to potentially inhibiting substrate binding.

To determine whether PLK-2 is important for meiotic progression independent of its role in homolog pairing and synapsis (*Harper et al., 2011*; *Labella et al., 2011*; *Sato-Carlton et al., 2017*), we used a degron-tagged allele (*Zhang et al., 2018*). Depletion of PLK-2 significantly delayed CHK-2 inactivation and CO designation (*Figure 3C, E*). We reasoned that if PLK-2 promotes timely CO designation primarily by inactivating CHK-2, then the delay observed in *plk-2* mutants should be rescued by depletion of CHK-2. Indeed, we found that AID-mediated depletion of CHK-2 restored earlier CO designation in *plk-2* mutants (*Figure 3D, F*).

As the SC assembles, PLK-2 is recruited to this structure by binding to a Polo box interacting motif on SYP-1 (*Sato-Carlton et al., 2017*; *Brandt et al., 2020*). Following CO designation at mid-pachytene, PLK-2 activity becomes restricted to the 'short arms' of each bivalent (*Sato-Carlton et al., 2017*). We speculated that recruitment of PLK-2 to the SC may promote its ability to inactivate CHK-2. To test this idea, we examined CHK-2 activity in *syp-1*$^{T452A}$ mutants, which lack the Polo box-binding motif (S-pT-P) that recruits Polo kinases to the SC (*Sato-Carlton et al., 2017*; *Figure 4A*). Consistent with prior analysis, this single-point mutation markedly delayed the appearance of bright COSA-1 foci (*Sato-Carlton et al., 2017*; *Figure 4B, D*). We found that CHK-2 activity was similarly extended in *syp-1*$^{T452A}$ mutants (*Figure 4B, D*). Depletion of CHK-2 restored earlier CO designation in *syp-1*$^{T452A}$ mutants (*Figure 4C, E*), as in *plk-2* mutants. Thus, binding of PLK-2 to the SC is important for downregulation of CHK-2, and conversely, the delay in CO designation in *syp-1*$^{T452A}$ mutants is a direct consequence of persistent CHK-2 activity.

We observed that CO designation was delayed to different extents by *plk-2* null mutations, induced depletion of PLK-2, and *syp-1*$^{T452A}$. We scored the timing of CO designation as the fraction of the length of the region of the gonad spanning meiotic prophase (prior to diplotene–diakinesis) in which nuclei display bright COSA-1 foci, since this ratio is fairly consistent for a given genotype/condition, more so than the absolute length of this zone or number of nuclei. The 'bright COSA-1 zone' was reduced from 55% of prophase in wild-type hermaphrodites to 44% in *plk-2* null mutants, 37% in PLK-2::AID+auxin, and 20% in syp-1(T452A) (*Figure 3C–F*; *Figure 4B, D*). The delay in *plk-2* null mutants is likely due in part to earlier prophase defects, including delayed pairing and synapsis (*Harper et al., 2011*; *Labella et al., 2011*). Depletion of PLK-2 caused less disruption of these early events (*Figure 4—figure supplement 1A*). SYP-1$^{T452A}$ caused even milder defects in pairing and synapsis (*Brandt et al., 2020*; *Sato-Carlton et al., 2017*), but a more dramatic delay in CHK-2 inactivation and CO designation (*Figures 3C–F and 4B, D*). These differences are likely due to the ability of PLK-1 to partially substitute for PLK-2 in inactivating CHK-2 (see below), while SYP-1$^{T452A}$ cannot recruit either

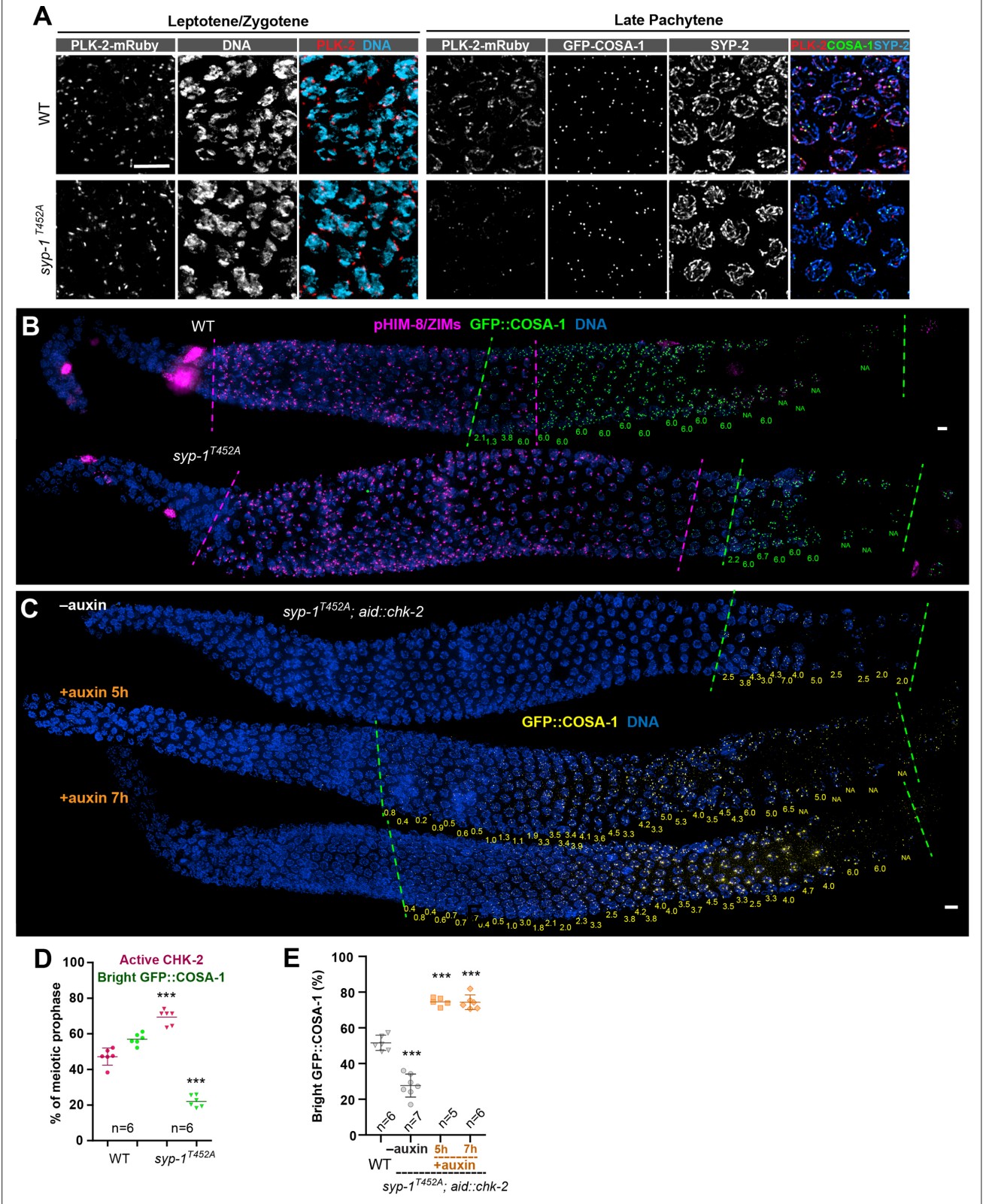

**Figure 4.** CHK-2 inactivation requires recruitment of Polo-like kinases to the synaptonemal complex (SC). (**A**) Representative leptotene/zygotene or late pachytene nuclei in wild-type or *syp-1^T452A* hermaphrodite gonads stained for PLK-2::mRuby (red), DNA (cyan), GFP::COSA-1 (green), and SYP-2 (blue). A conserved Polo box recruitment motif on the SC is absent in *syp-1^T452A* mutants. PLK-2 localized to pairing centers in leptotene/zygotene nuclei but failed to localize to SC in pachytene nuclei in *syp-1^T452A* mutants. (**B**) Representative hermaphrodite gonads stained for pHIM-8/ZIMs (magenta),

*Figure 4 continued on next page*

*Figure 4 continued*

GFP::COSA-1 (green), and DNA (blue), showing extension of CHK-2-active zone and delayed appearance of bright COSA-1 foci in *syp-1^T452A* mutants. Dashed magenta lines indicate the CHK-2-active zone, while green lines indicate the bright COSA-1 zone. The average number of bright COSA-1 foci per nuclei in each row is indicated in green below each image. Scale bars, 5 μm. (**C**) Representative hermaphrodite gonads stained for GFP::COSA-1 (yellow) and DNA (blue), showing that depletion of CHK-2 in *syp-1^T452A* mutants leads to early appearance of bright COSA-1 foci. Worms were treated with or without 1 mM auxin for 5 or 7 hr. Dash lines indicate the zone with bright COSA-1 foci. Scale bars, 5 μm. (**D**) Quantification of the extension of CHK-2-active zone and the delay in appearance of bright COSA-1 foci in worms as described in (**B**). *n* = number of gonads scored for each condition. ***p < 0.0001, two-sided Student's *t*-test. (**E**) Quantification of bright COSA-1 zone in worms maintained and treated as in (**C**). Wild-type worms were used as control. *n* = number of gonads scored for each condition. ***p < 0.0001, two-sided Student's *t*-test.

The online version of this article includes the following figure supplement(s) for figure 4:

**Figure supplement 1.** PLK-1 is not required to override the crossover assurance checkpoint in the absence of PLK-2.

**Figure supplement 2.** MPK-1 does not silence the crossover assurance checkpoint.

PLK-1 or PLK-2. This supports the conclusion that recruitment of Polo-like kinase to SCs is important for CHK-2 inactivation and CO designation.

Although mutation or depletion of PLK-2 or failure to recruit it to the SC delayed inactivation of CHK-2, markers for CHK-2 activity eventually disappeared and CO designation was detected in all these strains. We thus wondered how CHK-2 is inactivated in the absence of PLK-2. In *plk-2* null mutants, PLK-1 can partially substitute at Pairing Centers to promote pairing and synapsis (*Harper et al., 2011*; *Labella et al., 2011*). We tested whether PLK-1 might compensate for loss of PLK-2 in late prophase, as it does during early meiosis, by co-depleting both paralogs (*Figure 4—figure supplement 1A*). Compared to PLK-2 depletion alone, CO designation was further delayed when PLK-1 and PLK-2 were both depleted, indicating that PLK-1 can indeed contribute to CHK-2 inactivation during late prophase (*Figure 3C, E*; *Figure 4—figure supplement 1B, C*). However, CHK-2 was still eventually inactivated in the absence of both paralogs. We also tested whether the nonessential Plk1 homolog *plk-3* can promote CO designation. As previously reported (*Harper et al., 2011*), *plk-3* mutants showed no apparent meiotic defects, and we observed no further delay when we combined a *plk-3* deletion with depletion of PLK-1 and PLK-2 (data not shown). Thus, PLK-3 does not contribute to inactivating CHK-2, even in the absence of PLK-1 and PLK-2.

The ERK kinase MPK-1 promotes pachytene exit and has been proposed to be important for CO designation (*Church et al., 1995*; *Hayashi et al., 2007*; *Lee et al., 2007*; *Nadarajan et al., 2016*). We used the AID system to test whether MPK-1 promotes CHK-2 inactivation or CO designation. MPK-1::AID was undetectable in germ cells following 1 hr of auxin treatment (*Figure 4—figure supplement 2A*). Although depletion of MPK-1 led to a disordered appearance of the proximal germline, it did not cause any apparent delay in CO designation, either alone or in combination with *syp-1^T452A* (*Figure 4—figure supplement 2B–E*). We thus conclude that MPK-1 activity does not play a role in CO designation. We speculate that a rise in CDK activity and/or other spatially regulated signals in the proximal gonad may lead to CHK-2 inactivation even when Polo-like kinase activity is absent.

## Discussion

In summary, we find that CHK-2 activity is required to inhibit CO designation. CHK-2 is normally inactivated at mid-pachytene through the recruitment of PLK-2 to the SC and the formation of CO precursors, but delays in synapsis prevent the activation of PLK-2 and thereby prolong CHK-2 activity. Notably, delays in the establishment of CO intermediates also prolong CHK-2 activity, and thus full activation of PLK-2 may depend on the formation of CO intermediates on all chromosomes. Consistent with this idea, PLK-2-dependent phosphorylation of the central region protein SYP-4 (*Nadarajan et al., 2017*) increase markedly starting at mid-pachytene, concomitant with CO designation. How the formation of CO precursors promotes PLK-2 activity remains unclear, but the kinase does appear to associate with designated CO sites even when it cannot be recruited to the SC (*Zhang et al., 2018*).

Our data support the idea that PLK-2 directly regulates CHK-2 via inhibitory phosphorylation, but do not rule out the possibility that recruitment of PLK-2 to the SC leads indirectly to CHK-2 inactivation. A recent study reported that CHK-2 inactivation is reversible through mid-pachytene (*Castellano-Pozo et al., 2020*), consistent with the idea that it occurs through phosphorylation, which

can be reversed by phosphatase activity (*Kar and Hochwagen, 2021*). We speculate that phosphorylated CHK-2 may be degraded after mid-pachytene, resulting in irreversible inactivation.

Intriguingly, both CHK-2 and PLK-2 are active during leptotene/zygotene, when both kinases are bound to Pairing Centers (*Kim et al., 2015*; *Link et al., 2018*; *Penkner et al., 2009*; *Sato et al., 2009*; *Woglar et al., 2013*), implying that in this context PLK-2 does not inactivate CHK-2. We speculate that the configuration of CHK-2 and PLK-2 recruitment motifs on the Pairing Center proteins may make the target site(s) in the FHA domain of CHK-2 inaccessible to the active site of PLK-2. However, the activity of CHK-2 in early prophase also depends on an enigmatic meiotic regulator of PLK-2 activity, a heterodimeric complex of HAL-2 and HAL-3 (*Roelens et al., 2019*; *Zhang et al., 2012*). This may inhibit or antagonize phosphorylation of CHK-2. Further work will be needed to clarify how CHK-2 and PLK-2 work in concert at pairing centers during early meiosis, while PLK-2 antagonizes CHK-2 later in meiotic prophase.

In *C. elegans*, synapsis can occur normally in the absence of DSBs or recombination intermediates (*Dernburg et al., 1998*), and meiocytes surveil both synapsis and CO precursors to ensure chiasma formation and faithful segregation (reviewed by ). In organisms where synapsis depends on the stabilization of interhomolog joint molecules through the 'ZMM' pathway (*Pyatnitskaya et al., 2019*), cells may monitor synapsis as a proxy for the presence of CO-competent intermediates. However, in either case, SC assembly is essential for the surveillance. In mouse spermatocytes, PLK-1 localizes along SCs and promotes pachytene exit (*Jordan et al., 2012*). Thus, similar mechanisms may coordinate synapsis with meiotic progression in other organisms.

## Materials and methods

### Worm strains

All *C. elegans* strains were maintained on standard nematode growth medium (NGM) plates seeded with OP50 bacteria at 20°C. All epitope- and degron-tagged alleles analyzed in this study were fully functional, as indicating by their ability to support normal meiosis and development (*Supplementary file 1a*). Unless otherwise indicated, new alleles used in this study were generated by CRISPR/Cas9-mediated genome editing following a modified protocol as previously described (*Arribere et al., 2014*; *Paix et al., 2015*; *Zhang et al., 2018*).

See *Supplementary file 1b* for a list of new alleles generated in this study and *Supplementary file 1c* for reagents used to make these alleles. A list of worm strains used in this study is shown in *Supplementary file 1d*. Unless otherwise indicated, young adults (20–24 hr post-L4) were used for both immunofluorescence and western blot assays.

### Worm viability and fertility

To quantify brood sizes, male self-progeny, and embryonic viability, L4 hermaphrodites were picked onto individual seeded plates and transferred to new plates daily over 4 days. Eggs were counted daily. Viable progeny and males were scored when they reached the L4 or adult stages.

### Auxin-mediated protein depletion in worms

Auxin-mediated protein depletion was performed as previously described (*Guo et al., 2022*; *Zhang et al., 2015*). Briefly, worms were transferred to bacteria-seeded plates containing 1 mM indole-3-acetic acid (Acros Organics, Cat #122160250) and incubated for the indicated time periods before analysis.

### In vitro phosphorylation assay

Recombinant kinase-dead CHK-2 (CHK-2$^{KD}$, K199->R) was expressed and purified as described previously (*Kim et al., 2015*). For PLK-2, the full-length open reading frame was amplified from a *C. elegans* cDNA library and cloned into pFastBac1 (Life Technologies) with a GST tag at its N-terminus. GST-PLK-2 was expressed in insect Sf9 cells using the standard Bac-to-Bac system (Life Technologies) and then purified using glutathione Sepharose (GE Life Sciences).

In vitro kinase assays were performed in a buffer comprised of 25 mM HEPES pH 7.4, 50 mM NaCl, 2 mM EGTA, 5 mM MgSO$_4$, 1 mM DTT, and 0.5 mM NaF, supplemented with 0.5 mM Mg-ATP. 2 µM of GST-CHK-2$^{KD}$ were incubated with or without 0.2 µM of GST-PLK-2 at room temperature for 1 hr.

Kinase reactions were terminated by either quick freezing in liquid nitrogen or addition of sodium dodecyl sulfate (SDS) sample buffer. Proteins in SDS buffer were electrophoresed using gradient polyacrylamide gels (Genscript, #M00652). CHK-2 bands were excised and stored at 4°C. Proteins were then subjected to either in-solution or in-gel trypsin digestion, and phosphorylation sites were identified using mass spectrometry analyses (UC Davis).

## Mass spectrometry

An Xevo G2 QTof coupled to a nanoAcquity UPLC system (Waters, Milford, MA) was used for phosphorylation site identification. Briefly, samples were loaded onto a C18 Waters Trizaic nanotile of 85 μm × 100 mm; 1.7 μm (Waters, Milford, MA). The column temperature was set to 45°C with a flow rate of 0.45 ml/min. The mobile phase consisted of A (water containing 0.1% formic acid) and B (acetonitrile containing 0.1% formic acid). A linear gradient elution program was used: 0–40 min, 3–40% (B); 40–42 min, 40–85% (B); 42–46 min, 85% (B); 46–48 min, 85–3% (B); 48–60 min, 3% (B).

Mass spectrometry data were recorded for 60 min for each run and controlled by MassLynx 4.2 (Waters, Milford, MA). Acquisition mode was set to positive polarity under resolution mode. Mass range was set from 50 to 2000 Da. Capillary voltage was 3.5 kV, sampling cone at 25 V, and extraction cone at 2.5 V. Source temperature was held at 110°C. Cone gas was set to 25 l/hr, nano flow gas at 0.10 bar, and desolvation gas at 1200 l/hr. Leucine–enkephalin at 720 pmol/μl (Waters, Milford, MA) was used as the lock mass ion at $m/z$ 556.2771 and introduced at 1 μl/min at 45-s intervals with a 3 scan average and mass window of ±0.5 Da. The Mse data were acquired using two scan functions, corresponding to low energy for function 1 and high energy for function 2. Function 1 had collision energy at 6 V and function 2 had a collision energy ramp of 18–42 V.

RAW Mse files were processed using Protein Lynx Global Server (PLGS) version 3.0.3 (Waters, Milford, MA). Processing parameters consisted of a low energy threshold set at 200.0 counts, an elevated energy threshold set at 25.0 counts, and an intensity threshold set at 1500 counts. The databank used corresponded to *C. elegans* and was downloaded from uniprot.org and then randomized. Searches were performed with trypsin specificity and allowed for two missed cleavages. Possible structure modifications included for consideration were methionine oxidation, carbamidomethylation of cysteine, deamidiation of asparagine or glutamine, dehydration of serine or threonine, and phosphorylation of serine, threonine, or tyrosine.

## ALFA::CHK-2 immunoprecipitation

To obtain synchronized young adults, five to six L4 larvae animals were picked onto standard 60-mm culture plate spread with OP50 bacteria. Animals were grown at 20°C for 5 days until starved. 90 plates of L1 larvae for each genotype or condition were washed into 1.5-l liquid culture supplemented with HB101 bacteria. Worms were then grown with aeration at 200 rpm at 20°C for 3 days to reach adulthood. Six hours prior to harvest, 1 mM auxin or 0.25% ethanol (solvent control) was added to the liquid culture. Harvested worms were frozen in liquid nitrogen and stored at −80°C. The frozen worms were then processed using a prechilled Retsch mixer mill to break the cuticle, thawed on ice in cold lysis buffer (25 mM HEPES pH 7.4, 100 mM NaCl, 1 mM $MgCl_2$, 1 mM EGTA, 0.1% Triton X-100, 1 mM DTT, cOmplete protease inhibitors [Sigma #4693159001] and phosSTOP [Sigma #4906837001]). Lysates were processed using a Dounce homogenizer and sonicated using a Branson Digital Sonifier on ice and then centrifuged at 20,000 RCF for 25 min at 4°C. The supernatant was incubated with anti-ALFA selector (Nanotag Biotechnologies, #N1511) for 3 hr at 4°C. Beads were then washed six times with lysis buffer and three times with milli-Q water. Proteins on beads were then processed for phosphorylation site identification using mass spectrometry (UC Davis).

Briefly, beads were spun in a 10K MWCO filter (VWR, Radnor, PA) at room temperature for 10 min at 10,000 × $g$ and then washed with 50 mM ammonium bicarbonate. Beads were then subjected to reduction at 56°C for 45 min in 5.5 mM DTT followed by alkylation for 1 hr in the dark with iodoacetamide added to a final concentration of 10 mM. The beads were again washed with 50 mM ammonium bicarbonate followed by addition of sequencing grade trypsin to a final enzyme:substrate mass ratio of 1:50 and digested overnight at 37°C. Resultant peptides were then collected in a fresh clean centrifuge tube during a final spin of 16,000 × $g$ for 20 min. Peptides were dried down in a speed-vac and stored at −80°C. Prior to analysis, samples were reconstituted in 2% acetonitrile with 0.1% TFA. Samples were then loaded and analyzed as described above.

## Microscopy

Immunofluorescence experiments for *C. elegans* were performed as previously described (*Zhang et al., 2018*). Images shown in *Figure 4—figure supplement 2A* were obtained from worms dissected and fixed in the absence of Tween-20 to retain soluble proteins. Primary antibodies were obtained from commercial sources or have been previously described, and were diluted as follows: Rabbit anti-RAD-51 (1:5000, Novus Biologicals, #29480002), Rabbit anti-pHIM-8/ZIMs (1:500, *Kim et al., 2015*), Goat anti-SYP-1 (1:300, *Harper et al., 2011*), Rabbit anti-SYP-2 (1::500, *Colaiácovo et al., 2003*), Chicken anti-HTP-3 (1:500, *MacQueen et al., 2005*), Mouse anti-HA (1:400, Thermo Fisher, #26183), Mouse anti-GFP (1:500, Millipore Sigma, #11814460001), Mouse anti-FLAG (1:500, Sigma, #F1804), anti-ALFA-At647N (1:500, Nanotag Biotechnologies, N1502-At647N). Secondary antibodies labeled with Alexa 488, Cy3, or Cy5 were purchased from Jackson ImmunoResearch (WestGrove, PA) and used at 1:500. All images were acquired as z-stacks through 8–12 μm depth at z-intervals of 0.2 μm using a DeltaVision Elite microscope (GE) with a ×100, 1.4 N.A. or ×60, 1.42 N.A. oil-immersion objective. Iterative 3D deconvolution, image projection, and colorization were carried out using the softWoRx package and Adobe Photoshop CC 2021.

## Western blotting

Adult worms were picked into S-basal buffer and lysed by addition of SDS sample buffer, followed by boiling for 15 min with occasional vortexing. Whole worm lysates were then separated on 4–12% poly-acrylamide gradient gels (GenScript, #M00654), transferred to membranes, and blotted with mouse anti-HA (1:1000, Thermo Fisher, #26183), Guinea pig anti-HTP-3 (1:1500, *MacQueen et al., 2005*), mouse anti-α-tubulin (1:5000, Millipore Sigma, #05-829), rabbit anti-β-tubulin (1:2000, Abcam, ab6046), or HRP-conjugated anti-ALFA sdAb (1:1000, Nanotag Biotechnologies, N1505-HRP), HRP-conjugated anti-mouse secondary antibodies (Jackson Immunoresearch #115-035-068), HRP-conjugated anti-Rabbit secondary antibodies (Jackson Immunoresearch #111-035-144), HRP-conjugated anti-Guinea pig secondary antibodies (Jackson Immunoresearch #106-035-003). SuperSignal West Femto Maximum Sensitivity Substrate (Thermo Fisher, #34095) was used for detection.

## Timing of CHK-2 inactivation and CO designation

The CHK-2-active zone was determined by immunofluorescence with an antibody recognizing phosphorylated HIM-8 and ZIM proteins (*Kim et al., 2015*). Designated COs were detected using GFP::CO-SA-1. The lengths of regions corresponding to each meiotic stage vary among individual animals, but the length ratios of these regions are relatively consistent (*Kim et al., 2015*; *Stamper et al., 2013*). Thus, unless otherwise indicated, we quantified the CHK-2-active zone or bright COSA-1 zone as a ratio of the region showing positive staining to the total length of the region from meiotic onset to the end of pachytene. Meiotic onset was determined by the staining of meiosis-specific proteins SYP-1 and/or HTP-3.

Bright GFP::COSA-1 foci were identified using the intensity profile of GFP::COSA-1 throughout pachytene, as shown in *Figure 1D*. While early GFP::COSA-1 foci are relatively dim and sensitive to fixation and antibody staining conditions, COSA-1 foci at designated CO sites are relatively constant in intensity from their appearance until the end of pachytene. The absolute intensity of GFP::COSA-1 varied between samples, so it was not possible to identify designated CO sites using a fixed intensity threshold.

## Quantification and statistical analysis

Quantification methods and statistical parameters are described in the legend of each figure, including sample sizes, error calculations (standard deviation or standard error of the mean), statistical tests, and p values. p < 0.05 was considered to be significant.

## Acknowledgements

Some strains used in this work were provided by the *Caenorhabditis* Genetics Center (CGC), which is funded by NIH Office of Research Infrastructure Programs (P40 OD010440). We thank Yumi Kim for sharing the CHK-2 and PLK-2 plasmids, Andrew Ziesel, Nancy Hollingsworth, and members of the Dernburg lab for helpful discussions throughout this work.

Abby F Dernburg: National Institutes of Health (R01 GM065591) and Howard Hughes Medical Institute. The funders had no role in study design, data collection, and interpretation, or the decision to submit the work for publication.

## Additional information

### Funding

| Funder | Grant reference number | Author |
|---|---|---|
| Howard Hughes Medical Institute | | Abby F Dernburg |
| National Institutes of Health | R01 GM065591 | Abby F Dernburg |

The funders had no role in study design, data collection, and interpretation, or the decision to submit the work for publication.

### Author contributions

Liangyu Zhang, Conceptualization, Resources, Data curation, Software, Formal analysis, Supervision, Validation, Investigation, Visualization, Methodology, Writing - original draft, Project administration, Writing - review and editing; Weston T Stauffer, Investigation, Methodology, Writing - original draft, Writing - review and editing; John S Wang, Software, Validation, Investigation, Methodology, Writing - original draft, Writing - review and editing; Fan Wu, Investigation, Methodology; Zhouliang Yu, Chenshu Liu, Hyung Jun Kim, Investigation; Abby F Dernburg, Conceptualization, Supervision, Funding acquisition, Visualization, Writing - original draft, Project administration, Writing - review and editing

### Author ORCIDs

Liangyu Zhang ⓘ http://orcid.org/0000-0002-2701-0773
Weston T Stauffer ⓘ http://orcid.org/0000-0001-8998-5077
Abby F Dernburg ⓘ http://orcid.org/0000-0001-8037-1079

### Decision letter and Author response

Decision letter https://doi.org/10.7554/eLife.84492.sa1
Author response https://doi.org/10.7554/eLife.84492.sa2

## Additional files

### Supplementary files

• Supplementary file 1. This file includes four tables (*Supplementary file 1a-d*). Supplementary file 1a reports the viability and fertility of representative transgenic worm strains used in this study, which indicates that all epitope- and degron-tagged alleles support normal meiosis and development. Supplementary file 1b lists the worm alleles generated in this study. Supplementary file 1c lists the crRNA, repair templates, and genotyping primers generated in this study. Supplementary file 1d lists the worm strains used in this study.

• MDAR checklist

### Data availability

All data generated or analysed during this study are included in the manuscript and supporting file; Source Data files have been provided for Figure 3—figure supplement 1 and Figure 3—figure supplement 2.

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

## Appendix 1

### Appendix 1—key resources table

| Reagent type (species) or resource | Designation | Source or reference | Identifiers | Additional information |
|---|---|---|---|---|
| Gene (*Caenorhabditis elegans*) | *chk-2* | WormBase | Wormbase ID: WBGene00000499 | |
| Gene (*Caenorhabditis elegans*) | *plk-2* | WormBase | Wormbase ID: WBGene00004043 | |
| Gene (*Caenorhabditis elegans*) | *plk-1* | WormBase | WormBase ID: WBGene00004042 | |
| Gene (*Caenorhabditis elegans*) | *mpk-1* | WormBase | WormBase ID: WBGene00003401 | |
| Gene (*Caenorhabditis elegans*) | *cosa-1* | WormBase | WormBase ID: WBGene00022172 | |
| Gene (*Caenorhabditis elegans*) | *syp-1* | WormBase | WormBase ID: WBGene00006375 | |
| Strain, strain background (*Escherichia coli*) | OP50 | *Caenorhabditis* Genetics Center (CGC) | N/A | |
| Strain, strain background (*Escherichia coli*) | DH10Bac | Thermo Fisher Scientific | Cat. #10361012 | Competent *E. coli* |
| Strain, strain background (*Caenorhabditis elegans*) | For *C. elegans* allele and strain information, see **Supplementary file 1b, d** | This paper | N/A | Strains are available in Abby Dernburg's lab |
| Genetic reagent (*Caenorhabditis elegans*) | For *C. elegans* mutations, see **Supplementary file 1b, d** | This paper | N/A | Mutations are available in Abby Dernburg's lab |
| Cell line (*Spodoptera frugiperda*) | Sf9 insect cells | Thermo Fisher Scientific | Cat. #11496015 | |
| Antibody | anti-SYP-1 (Goat polyclonal) | (**Harper et al., 2011**) PMID: 22018922 | N/A | IF (1:300) |
| Antibody | anti-RAD-51 (Rabbit polyclonal) | Novus Biologicals | Cat. #29480002; RRID:AB_2284913 | IF (1:5000) |
| Antibody | anti-pHIM-8/ZIMs (Rabbit polyclonal) | (**Kim et al., 2015**) PMID: 26506311 | N/A | IF (1:500) |
| Antibody | anti-SYP-2 (Rabbit polyclonal) | (**Colaiácovo et al., 2003**) PMID: 12967565 | N/A | IF (1:500) |
| Antibody | anti-β-tubulin (Rabbit polyclonal) | Abcam | Cat. #ab6046; RRID:AB_2210370 | WB (1:2000) |
| Antibody | anti-HTP-3 (Chicken polyclonal) | (**MacQueen et al., 2005**) PMID: 16360034 | N/A | IF (1:500) |
| Antibody | anti-HTP-3 (Guinea pig polyclonal) | (**MacQueen et al., 2005**) PMID: 16360034 | N/A | WB (1:500) |
| Antibody | anti-α-tubulin (Mouse monoclonal) | Millipore Sigma | Cat. #05-829; RRID:AB_310035 | WB (1:5000) |
| Antibody | anti-HA (Mouse monoclonal) | Thermo Fisher Scientific | Cat. #26183; RRID:AB_10978021 | IF (1:400), WB (1:1000) |
| Antibody | anti-GFP (Mouse monoclonal) | Millipore Sigma | Cat. #11814460001; RRID:AB_390913 | IF (1:500) |

*Appendix 1 Continued on next page*

*Appendix 1 Continued*

| Reagent type (species) or resource | Designation | Source or reference | Identifiers | Additional information |
|---|---|---|---|---|
| Antibody | anti-FLAG (Mouse monoclonal) | Millipore Sigma | Cat. #F1804; RRID:AB_262044 | IF (1:500) |
| Antibody | anti-ALFA-At647N (Alpaca monoclonal, Nanobody clone 1G5 produced in *E. coli*) | Nanotag Biotechnologies | Cat. #N1502-At647N | IF (1:500) |
| Antibody | HRP-conjugated anti-ALFA sdAb (Alpaca monoclonal, Nanobody clone 1G5 produced in *E. coli*) | Nanotag Biotechnologies | Cat. #N1505-HRP | WB (1:1000) |
| Recombinant DNA reagent | pFastBac1 GST-CHK-2$^{KD}$ (plasmid) | (*Kim et al., 2015*) PMID: 26506311; This paper | N/A | Generously provided by Yumi Kim |
| Recombinant DNA reagent | pFastBac1 GST-PLK-2 (plasmid) | This paper | N/A | Generously provided by Yumi Kim |
| Sequence-based reagent | CRISPR tracrRNA | Integrated DNA Technologies | Cat. #1072534 | |
| Sequence-based reagent | dpy-10 crRNA | (*Arribere et al., 2014*) PMID: 25161212; Integrated DNA Technologies | N/A | 5'-GCUACCAUAGGC ACCACGAG-3' |
| Sequence-based reagent | dpy-10 (cn64) repair template | (*Arribere et al., 2014*) PMID: 25161212; Integrated DNA Technologies | N/A | Oligo: 5'-CACTTGAA CTTCAATACGGCAAGA TGAGAATGACTGGAAA CCGTACCGCATGCGGT GCCTATGGTAGCGGAG CTTCACATGGCTTCAG ACCAACAGCCTAT-3' |
| Sequence-based reagent | crRNAs, repair templates and genotyping primers | This paper | N/A | *Supplementary file 1c* |
| Peptide, recombinant protein | *S. pyogenes* Cas9-NLS purified protein | QB3 MacroLab at UC Berkeley | N/A | |
| Peptide, recombinant protein | ALFA selector | Nanotag Biotechnologies | Cat. #N1511 | |
| Peptide, recombinant protein | Glutathione Sepharose | GE Life Sciences | Cat. #17-5132-01 | |
| Commercial assay or kit | SuperSignal West Femto Maximum Sensitivity Substrate kit | Thermo Fisher Scientific | Cat. #34095 | |
| Chemical compound, drug | Auxin, indole-3-acetic acid | Acros Organics | Cat. #122160250 | |
| Chemical compound, drug | DAPI (4',6-Diamidino-2-phenylindole dihydrochloride) | Thermo Fisher Scientific | Cat. #62247 | |
| Software, algorithm | SoftWorx package | Applied Precision; GE Healthcare Bio-Sciences | http://www.sussex.ac.uk/gdsc/intranet/pdfs/softWoRx%20user%20manual | |
| Software, algorithm | ImageJ | NIH | https://imagej.nih.gov/ij | |
| Software, algorithm | Adobe Photoshop 2021 | Adobe Systems | https://www.adobe.com | |

*Appendix 1 Continued on next page*

*Appendix 1 Continued*

| Reagent type (species) or resource | Designation | Source or reference | Identifiers | Additional information |
|---|---|---|---|---|
| Software, algorithm | Adobe Illustrator 2021 | Adobe Systems | https://www.adobe.com | |
| Software, algorithm | T-COFFEE | Swiss Institute of Bioinformatics (**Notredame et al., 2000**) PMID:10964570 | http://tcoffee.vital-it.ch/apps/tcoffee | |
| Software, algorithm | IBS_1.0.1 | (**Liu et al., 2015**) PMID: 26069263 | http://ibs.biocuckoo.org/download.php | |
| Software, algorithm | GraphPad Prism | GraphPad Software, Inc | http://www.graphpad.com | |
| Software, algorithm | Protein Lynx Global Server (PLGS) version 3.0.3 | Waters | https://www.waters.com/waters/en_US/ProteinLynx-Global-SERVER-(PLGS)/nav.htm?cid = 513,821 | |
| Other | Polyacrylamide gels (10 wells) | Genscript | Cat. #M00652 | Protein electrophoresis, see 'Materials and methods' section in the paper for details |
| Other | Polyacrylamide gels (15 wells) | Genscript | Cat. #M00654 | Protein electrophoresis, see 'Materials and methods' section in the paper for details |
| Other | SlowFade Glass Antifade Mountant | Thermo Fisher Scientific | Cat. #S36917 | See 'Materials and methods' section in the paper for details |

