## [Editor Report]

Zhang et al. present convincing data describing a role for Polo-like kinase PLK-2 in restricting the activity of Chk2 kinase and coordinating synapsis of homologous chromosomes with the progression of meiotic prophase in *C. elegans*. By revealing PLK-2-dependent and -independent mechanisms of CHK-2 activity, this work provides a valuable understanding of the major regulators of meiotic progression.

---

## [Decision Letter]

**Decision letter after peer review:**

Thank you for submitting your article "Recruitment of Polo-like kinase couples synapsis to meiotic progression via inactivation of CHK-2" for consideration by *eLife*. Your article has been reviewed by 3 peer reviewers, and the evaluation has been overseen by a Reviewing Editor and Jessica Tyler as the Senior Editor. The following individual involved in the review of your submission has agreed to reveal their identity: Jeff Sekelsky (Reviewer #1).

Essential revisions:

The two main points highlighted below relate to the CHK-2 phospho-mutants and to the effect of the presence of the kinases as opposed to the requirement of their kinase activity. Most experiments we discussed presented complications and we therefore will not require them for publication. We would in turn 'suggest' them as a means to enhance the significance of the findings, if possible.

1) Did you consider expressing the different CHK-2 mutants from heterologous loci, all in the background of endogenous AID::CHK-2? Alternatively, providing wild-type AID::CHK-2 from an alternative locus could be an option. We do appreciate that given the stability effect of the phospho-mutants, any of these strategies might not be helpful.

2) Some attempts to differentiate between the requirement of the kinases vs the requirement of their kinase activity would be highly beneficial for the manuscript. Once again, given that these kinases play roles earlier in meiosis, the use of kinase-dead alleles might not be possible. Have the authors tried an analog-sensitive strategy to inhibit kinase activity?

Additionally, please go over the reviewers' comments carefully as some of them point out that some claims might need rewording.

*Reviewer #1 (Recommendations for the authors):*

It is worth clarifying in the Results section how long it takes for a nucleus to progress through prophase I (i.e., was the protein depleted for the entirety of meiosis up to the point that the nuclei were imaged, or was the protein only depleted for part of prophase I?).

1:4 – "CHK-2-dependent phosphorylation sites on HIM-8 and ZIM-1/2/3 at Pairing Centers" – I'm assuming this means this happens exclusively at pairing centers and not that the antibody only works at pairing centers, but the wording could imply the latter.

11:1 – Typo ("paring" should be "pairing").

Figure 1:

The meaning of the axis label "% of bright GFP::COSA-1 zone" is unclear before carefully reading the figure legend. It would be best to additionally state what this metric represents in the actual body of the text.

In Figure 1D, the bright green of GFP::COSA-1 in the graph is a bit challenging to read.

Figure 2

The color of SYP-1 (or at least the color of the text-the pseudocoloring of the microscopy appears to be magenta) should be changed to magenta to make the figure accessible to colorblind readers.

Figure 2-S1:

Change the color of RAD-51 to magenta.

Figure 3-S1:

The text size of the axes and labels of the mass spec graph could be increased to improve legibility.

Figure 4-S1 and associated Methods text:

Were the gain and/or brightness and contrast modified to account for the intense pHIM-8/ZIMs staining? If so, it would be informative to include this in the Microscopy methods.

*Reviewer #3 (Recommendations for the authors):*

Figure 2 legend: "Dashed green lines indicate the earliest nuclei with bright COSA-1 foci and the end of pachytene, respectively." – please clarify that the "respectively" refers to the left line and the right line.

p7 lines 14,16: Is there a reference for high versus intermediate levels of CHK-2 activity in these backgrounds? (The Kim et al. 2015 paper shows similar lengths of CHK-2 active zones in him-8 and him-5 mutants.) Otherwise, this should be marked as speculation.

p. 9 line 5: The authors speculate that phosphorylation of CHK-2 T120 leads to CHK-2 degradation, which is plausible given their results; the unexpected finding that blocking this phosphorylation with mutation to alanine (Figure 3 sup. 2) also apparently leads to instability should be discussed.

p. 9: the non-phosphorylatable alleles of CHK-2 turned out to be loss-of-function alleles, so the specific questions of whether phosphorylation of T120 or S116 leads to inactivation could not be asked in this work. Have the authors considered providing wild-type AID:chk-2 at a heterologous locus in these mutant backgrounds, to assess these alleles after depletion (after CHK-2 has performed its earlier function)?

p. 10, lines 20-21: the quantitative difference in CHK-2 activity between the depletion of PLK-2 and SYP-1(T452A) mutants is striking and somewhat counterintuitive since PLK-2 presumably at least exists in the SYP-1 phosphomutant. Is it possible that the difference lies in wild-type SYP-1's ability to recruit PLK-1? This could be experimentally addressed or discussed.

p. 12, line 19: The reference Sato-Carlton et al. 2020 shows that HIM-3 is phosphorylated independently of PLK-2 and PLK-1; this is opposite to what is stated here.

p. 11, line 3: SYP-2 should be corrected to SYP-1.

p. 17, line 4: correct "uniport" → "uniprot".

---

## [Author Response]

Essential revisions:The two main points highlighted below relate to the CHK-2 phospho-mutants and to the effect of the presence of the kinases as opposed to the requirement of their kinase activity. Most experiments we discussed presented complications and we therefore will not require them for publication. We would in turn 'suggest' them as a means to enhance the significance of the findings, if possible.1) Did you consider expressing the different CHK-2 mutants from heterologous loci, all in the background of endogenous AID::CHK-2? Alternatively, providing wild-type AID::CHK-2 from an alternative locus could be an option. We do appreciate that given the stability effect of the phospho-mutants, any of these strategies might not be helpful.

We recognize that the instability we observed for CHK-2 mutant proteins makes their analysis less than fully satisfying. We considered various ways to circumvent this issue, including expressing these mutant alleles in the presence of a degradable wild-type copy (as suggested above). However, as the reviewers/editors have noted, such experiments can be technically challenging. One barrier is the challenges associated with expression of transgenes in the germline of *C. elegans*. While methods for single-copy insertion (e.g., MosSCI or Cas9-based editing) have enabled germline expression of transgenes, this often fails to recapitulate the expression levels of endogenous genes. This is usually attributed to the presence of multiple potent and silencing pathways (including RNAi and piRNAs) in the germline. It is particularly daunting to introduce *chk-2* transgenes, since *chk-2* is an unusually large gene for *C. elegans* (~8 kb), well beyond the typical size of insertions made with CRISPR/Cas9, MosSCI, or other available transgenic methods. Moreover, the endogenous *chk-2* is on a chromosome arm; such distally positioned genes are typically protected from germline silencing by mechanisms that remain poorly understood, further complicating efforts to design and introduce additional copies. Thus, we feel that the effort required to overcome the technical barriers to such experiments outweighs any potential information that they would yield.

2) Some attempts to differentiate between the requirement of the kinases vs the requirement of their kinase activity would be highly beneficial for the manuscript. Once again, given that these kinases play roles earlier in meiosis, the use of kinase-dead alleles might not be possible. Have the authors tried an analog-sensitive strategy to inhibit kinase activity?

We appreciate this comment, as well as the reviewers’ recognition (again) that such experiments might be technically prohibitive. We agree that it would potentially be helpful to determine whether the role of PLK-2 and/or CHK-2 in promoting crossover designation can be separated from their kinase activities. We have engineered analog-sensitive alleles to inhibit PLK-1 and PLK-2 (as well as CDK-2) in other contexts, but this approach was not useful here for 2 reasons: (1) the degree of inhibition was somewhat variable, and more crucially, (2) the kinetics of inhibition were not rapid enough to allow for activity of the Polo-like kinases during early meiosis and inhibition shortly thereafter. The activity of most small molecules, including ATP analogs, in adult *C. elegans* is severely limited by the low permeability of the worm cuticle and/or the activity of efflux pumps. We also found that CHK-2 is refractory to the analog-sensitive approach; our efforts to engineer such alleles resulted in severe loss of kinase activity in the absence of inhibitors.

Auxin (indole acetic acid) is unusually small and water-soluble, and thus diffuses relatively quickly into the germline. It often induces robust protein degradation in ~1 hour, and this approach enabled us to induce degradation of CHK-2 and PLK-1/2 far more quickly than we could inhibit the kinase activity of PLK-1/2 using ATP analogs. Even this approach is limited by diffusion; we know that auxin-induced degradation can occur markedly faster in other experimental contexts, including in oocytes and embryos freshly dissected from adult *C. elegans*.

Thus, we do not have suitable tools to address this issue beyond what we have already included in the manuscript.

Additionally, please go over the reviewers' comments carefully as some of them point out that some claims might need rewording.Reviewer #1 (Recommendations for the authors):It is worth clarifying in the Results section how long it takes for a nucleus to progress through prophase I (i.e., was the protein depleted for the entirety of meiosis up to the point that the nuclei were imaged, or was the protein only depleted for part of prophase I?)

We have now addressed this in the figure legend of Figure 1A. Cells enter meiosis in the distal “arms” of the gonad and move proximally towards the spermatheca and uterus at a velocity of about one cell diameter per hour (Deshong et al., 2014). This should be regarded as a rough approximation and varies over the reproductive lifespan of adult hermaphrodites.

1:4 – "CHK-2-dependent phosphorylation sites on HIM-8 and ZIM-1/2/3 at Pairing Centers" – I'm assuming this means this happens exclusively at pairing centers and not that the antibody only works at pairing centers, but the wording could imply the latter.

We thank the reviewer for pointing out this. We have revised the comment as follow:

“CHK-2-dependent phosphorylation sites on the four paralogous pairing center proteins, HIM-8 and ZIM-1/2/3…”.

11:1 – Typo ("paring" should be "pairing").

Thanks for noting this; we have corrected the mistake.

Figure 1:The meaning of the axis label "% of bright GFP::COSA-1 zone" is unclear before carefully reading the figure legend. It would be best to additionally state what this metric represents in the actual body of the text.

We understand that for a ratio, it is more readable if both the numerator and the denominator are shown on a graph. But we feel it is not easy to precisely describe the denominator using simple words without causing confusion. So, we prefer just to indicate that the axis is a ratio on the graph and rely on the readers to read the figure legend if they need the detailed information.

In Figure 1D, the bright green of GFP::COSA-1 in the graph is a bit challenging to read.

We have changed the color to grey to improve readability.

Figure 2The color of SYP-1 (or at least the color of the text-the pseudocoloring of the microscopy appears to be magenta) should be changed to magenta to make the figure accessible to colorblind readers.

Thanks for noting this; we have made the suggested change.

Figure 2-S1:Change the color of RAD-51 to magenta.

We have changed the color of RAD-51 to magenta.

Figure 3-S1:The text size of the axes and labels of the mass spec graph could be increased to improve legibility.

We have made these changes.

Figure 4-S1 and associated Methods text:Were the gain and/or brightness and contrast modified to account for the intense pHIM-8/ZIMs staining? If so, it would be informative to include this in the Microscopy methods.

We thank the reviewer for this question. We think the reviewer is referring to the difference of pHIM-8/ZIMs intensity in the pre-meiotic nuclei in control and PLK-depleted worms. The intense signal in pre-meiotic nuclei does not correspond to HIM-8/ZIMs, but instead to an unknown epitope recognized by the antibody in mitotic cells (Kim et al., 2015). The intensity of this pre-meiotic epitope is greatly elevated by PLK-1/2 depletion, presumably due to arrest in mitosis. We adjusted the brightness in this figure to reveal the pHIM-8/ZIMs signal in meiotic nuclei, but the background intensity outside meiotic nuclei were kept the same in control and PLK-depleted worms, allowing comparison of signal intensity between different conditions. We have clarified this in the revised manuscript.

Reviewer #3 (Recommendations for the authors):Figure 2 legend: "Dashed green lines indicate the earliest nuclei with bright COSA-1 foci and the end of pachytene, respectively." – please clarify that the "respectively" refers to the left line and the right line.

We thank the review for this suggestion. We have changed the statement to “Dashed green lines on the left and right indicate the earliest nuclei with bright COSA-1 foci and the end of pachytene, respectively.”

p7 lines 14,16: Is there a reference for high versus intermediate levels of CHK-2 activity in these backgrounds? (The Kim et al. 2015 paper shows similar lengths of CHK-2 active zones in him-8 and him-5 mutants.) Otherwise, this should be marked as speculation.

We have now elaborated our description of the differences seen in synapsis- versus DSB/recombination-deficient mutants. As reported in Kim et al. (2015) and elsewhere, synapsis defects prolong the clustered chromosome morphology (sometimes described as a “crescent-shaped” DAPI-staining mass) characteristic of the leptotene-zygotene stage, while defects in DSB induction or recombination result in an extended “early pachytene” stage, with chromosomes distributed around the nuclear periphery. More and brighter foci of phosphorylated HIM-8/ZIM proteins are seen in transition-zone like nuclei than in early pachytene nuclei, indicating that the level of CHK-2 activity, at least at the Pairing Centers, is higher in leptotene/zygotene than in early pachytene.

p. 9 line 5: The authors speculate that phosphorylation of CHK-2 T120 leads to CHK-2 degradation, which is plausible given their results; the unexpected finding that blocking this phosphorylation with mutation to alanine (Figure 3 sup. 2) also apparently leads to instability should be discussed.

We have discussed this in the revision.

p. 9: the non-phosphorylatable alleles of CHK-2 turned out to be loss-of-function alleles, so the specific questions of whether phosphorylation of T120 or S116 leads to inactivation could not be asked in this work. Have the authors considered providing wild-type AID:chk-2 at a heterologous locus in these mutant backgrounds, to assess these alleles after depletion (after CHK-2 has performed its earlier function)?

Please see our response to this suggestion above.

p. 10, lines 20-21: the quantitative difference in CHK-2 activity between the depletion of PLK-2 and SYP-1(T452A) mutants is striking and somewhat counterintuitive since PLK-2 presumably at least exists in the SYP-1 phosphomutant. Is it possible that the difference lies in wild-type SYP-1's ability to recruit PLK-1? This could be experimentally addressed or discussed.

Yes, we believe that phosphorylated SYP-1 may recruit a small amount of PLK-1, especially in the absence of PLK-2. We have discussed this in the revision.

p. 12, line 19: The reference Sato-Carlton et al. 2020 shows that HIM-3 is phosphorylated independently of PLK-2 and PLK-1; this is opposite to what is stated here.

We have corrected this mistake.

p. 11, line 3: SYP-2 should be corrected to SYP-1.

We have corrected SYP-2 to SYP-1.

p. 17, line 4: correct "uniport" → "uniprot".

We have corrected this typo in the revision.